## RESEARCH ARTICLE

# When access ≠ acceptance: How clinical specialty demands shape mutual recognition on medical examination/test result reuse in Chinese hospitals: A study on a pilot province of China's medical digital reform

Chao Song[1,2], Xinmian Huang[2], Shasha Qian[2], Chaoyun Yuan[3], Shuning Liu[4], Jun Zhou[5]*

1 Laboratory Medicine Center, Zhejiang Center for Clinical Laboratories, Zhejiang Provincial People's Hospital (Affiliated People's Hospital), Hangzhou Medical College, Hangzhou, Zhejiang, China, 2 Department of Medical Affairs, Zhejiang Provincial People's Hospital (Affiliated People's Hospital), Hangzhou Medical College, Hangzhou, Zhejiang, China, 3 Department of Information Center, Zhejiang Provincial People's Hospital (Affiliated People's Hospital), Hangzhou Medical College, Hangzhou, Zhejiang, China, 4 Department of International and Domestic Cooperation, Zhejiang Provincial People's Hospital (Affiliated People's Hospital), Hangzhou Medical College, Hangzhou, Zhejiang, China, 5 Department of Laboratory Medicine, Hangzhou Children's Hospital, Hangzhou, Zhejiang, China

* zhoujun_zju@outlook.com

## Abstract

### Background

Despite widespread electronic health records adoption, interoperability for sharing examination/test results across healthcare institutions remains limited, leading to redundant testing, increased costs, and compromised care. China's mutual recognition policy for medical examination/test results, implemented via the interoperable results sharing platform(IRSP), aims to address this. However, variations in adoption across clinical specialties and the impact of hospital-level pathway interventions are poorly understood.

### Methods

Utilizing hospital-level administrative data from Zhejiang Province's IRSP (Oct 2023 – Sep 2024), this quasi-experimental study compared three intervention hospitals (blocking the "Overlook Access" pathway) with three control hospitals. We analyzed core recognition metrics (Access Rate-AR, Total Recognition Rate-TRR, Cross-Hospital Access Rate-CHAR, Cross-Hospital Recognition Rate-CHRR) across 12 clinical specialties. Analyses employed magnitude-based inference for intervention effects, Spearman correlations for specialty variations, and descriptive statistics for hospital-type comparisons.

**Data availability statement:** All relevant data are within the paper and its Supporting Information files.

**Funding:** The authors received financial support for this work from the following projects: 1) Soft Science Research Project of Zhejiang Provincial Department of Science and Technology (Grant Number: 2024C35061, received by CS, SL, and CY); 2) Medical Science and Technology Project of Zhejiang Province (Grant Number: 2023KY047, received by CS, SQ, and CY). The funders had no role in study design, data collection and analysis, decision to publish, or preparation of the manuscript.

**Competing interests:** The authors have declared that no competing interests exist.

## Results

Blocking the "Overlook Access" pathway significantly increased access metrics (AR: Intervention median 98.9% vs. Control 44.6%, Cohen's d = 2.02; CHAR: 99.5% vs. 57.6%, d = 2.85) but paradoxically decreased TRR (18.2% vs. 44.3%, d = −2.72), with minimal impact on CHRR. Substantial variations existed across specialties: Orthopedics and traditional Chinese medicine showed consistently higher access and recognition, while hepatobiliary and endocrinology faced significant challenges. Pediatrics exhibited high access but critically low recognition (e.g., Hospital H: TRR 2.05%, CHRR 2.82%), attributed to rapid physiological changes and data applicability concerns. Strong correlations existed within access metrics (AR-CHAR, ρ = 0.92, p < 0.001) and within recognition metrics (TRR-CHRR, ρ = 0.88, p < 0.001), but weak correlations between access and recognition.

## Conclusion

This study reveals a critical distinction between access to external medical records and their actual clinical recognition, demonstrating that information interventions alone are insufficient to improve the recognition rates. Clinical specialty-specific factors significantly influence recognition behaviors, reflecting variations in data utility, stability, and diagnostic practices. Institutional success in promoting mutual recognition depends on comprehensive, multi-level strategies. The IRSP exemplifies China's progress in health data interoperability, yet sustainable mutual recognition ultimately hinges on clinical relevance rather than mere accessibility.

## Introduction

Electronic Health Records systems (EHRs) have revolutionized the capture of clinical data [1,2]; however, their siloed implementation restricts cross-institutional care coordination. Interoperability, defined as the capacity of diverse systems to exchange and interpret shared data, is crucial for facilitating a seamless flow of patient information across institutions [3–5]. Health Information Exchange (HIE), which refers to the electronic sharing of clinical information across the boundaries of healthcare organizations, has been promoted as a significant technological application in medicine. It has the potential to enhance the efficiency, cost-effectiveness, quality, and safety of healthcare delivery [6–8]. Statistics indicated that despite a high adoption rate of EHRs, only 36% of office-based providers reported transmitting any electronic data to external providers in 2017 [9,10]. A web-based survey of 320 ambulatory care providers in Illinois suggested that governments could further enhance the usage rate of HIE by promoting large-scale interoperability efforts, improving external information technology support, and redesigning adaptable workflows [11]. In 2018, approximately one-third of patients who visited a physician reported experiencing a breakdown in information exchange, which included the necessity to repeat a laboratory test or provide imaging results [12]. This fragmentation resulted in redundant tests

and examinations because healthcare providers often lack access to a patient's complete medical history [13]. In 2012, a study indicated that HIE could reduce duplicative testing by 39%, which helps lower inspection and testing costs [14]. The financial implications of limited interoperability are significant. The inability to access or share patient records across institutions leads to repetitive tests and examinations, which substantially contribute to healthcare costs.

In recent years, with the advancement of the Diagnosis-Related Groups (DRG) payment system in China's healthcare system, the allocation of medical resources and cost control have become critical issues in health administration [15–17]. Against this backdrop, the mutual recognition policy for medical examination/test results (MRP-MER) has been widely implemented as a key measure to optimize resource utilization and to improve the quality and efficiency of healthcare services in China. The MRP-MER is defined as a policy mechanism aimed at reducing redundant testing by enabling the cross-institutional acceptance of diagnostic results. Zhejiang Province, with its provincial capital Hangzhou, is recognized as a national pilot province in the MRP-MER, playing a significant role in the Yangtze River Economic Belt in China [18]. The interoperable results sharing platform (IRSP), established in 2021 with the Chinese name Zhejiang Medical Mutual Recognition, aims to achieve the MRP-MER by leveraging a province-wide integrated intelligent healthcare data infrastructure. While similar to HIE, IRSP is distinguished by its emphasis on the cross-hospital delivery of patients' examination and test data reports, facilitating rapid sharing among hospitals. Within the hierarchical medical system in China [19], IRSP enables patients to grant physicians at various hospitals access to their prior examination and test results, thereby reducing unnecessary duplicate examinations and tests [20]. Such integration is critical for improving the continuity of care, reducing redundant tests, and enhancing the efficiency of healthcare delivery [21]. The platform's capability to integrate data from multiple sources, including laboratory test reports and medical images, ensures that healthcare providers obtain a comprehensive view of a patient's health status. This integration enhances diagnostic accuracy and improves treatment outcomes [22]. As of the end of 2024, the IRSP had been operational for three years. Hospitals in the pilot regions had become familiar with the functionalities of the IRSP and physicians had integrated its use into their clinical workflows. To facilitate the broader promotion and application of this government-led policy, relevant performance metrics had been formulated with a focus on the IRSP. In pursuit of superior performance metrics, certain hospitals had intervened in physicians' access behaviors by modifying in-hospital IRSP pathways.

The mutual recognition of medical examination and laboratory results plays a significant role in optimizing the allocation of medical resources and improving service efficiency in China. However, its practical implementation faces multiple challenges, including behavioral motivation mechanisms, differences among clinical specialties, variations in management strategies, and the assurance of data quality. Clinical departments differ in their needs and requirements for cross-institutional mutual recognition of examination and test results; nevertheless, specific research addressing these differences remains limited. In-depth analysis of the complex relationship between information access and mutual recognition behaviors, as well as clarification of the multifaceted factors influencing mutual recognition, are not only essential for increasing mutual recognition rates and refining the tiered healthcare delivery system, but also provide a scientific foundation for future policy formulation and health information system development.

## Method

### Data source and setting

The study utilized aggregated, hospital-level administrative data from the Zhejiang Province IRSP covering the period from October 1, 2023, to September 30, 2024. This full-year period was selected to mitigate seasonal variations and ensure a representative sampling of physician behavior patterns. The IRSP recorded daily cross-hospital examination and test recognition events and calculated key performance metrics, such as recognition rates and access rates. Hospitals accessed these anonymized, aggregated metrics through the IRSP's statistical module for internal management optimization. Data underwent source-level anonymization by the IRSP prior to researcher access, ensuring that no individual

patient identifiers were available. Ethical approval was granted with a waiver of individual informed consent, as the study involved only aggregated, non-identifiable data that serves significant public health interests without individual risk.

## Hospitals selection and characteristics

Hospitals were purposively selected to ensure representation of the dominant types of China's tiered healthcare system. This selection included four General Hospitals (designated A, B, C, D), two Traditional Chinese Medicine (TCM) Hospitals (E, F), and two Specialty Hospitals (G: Cancer Hospital, H: Children's Hospital). All institutions met the operational scale criterion of exceeding ¥1.6 billion (approximately $225 million USD) in annual medical revenue during 2023 [23]. Geographically concentrated in the provincial pilot capital of Hangzhou, this cohort collectively manages over 70% of the city's provincial-level medical service volume [23]. This concentration allows for standardized policy exposure and provides insights derived from high-volume, resource-rich tertiary settings. During the study period, the eight large hospitals received a total of 26,759,077 delivered reports through the IRSP, which included 3,639,368 precision-delivered reports specifically sent to fulfill physicians' orders. Additionally, there were 3,265,038 instances of physicians accessing reports, alongside 2,449,006 instances where physicians explicitly made recognition decisions. Furthermore, there were 1,190,431 instances where reports were overlooked and not accessed.

## Study design and intervention

The standard IRSP workflow initiates when a physician orders an examination or test, triggering an automated query for existing relevant reports from both local and external hospitals. When matching reports are identified, the system delivers them to the local EHRs accompanied by a physician review prompt. The standard IRSP workflow is shown in Fig 1. A physician must then sequentially: (1) accesses the delivered reports, and (2) makes a formal recognition decision ('Recognize' or 'Not Recognize'). Recognition results in cancellation of the new order, while non-recognition permits order execution in the local hospital. The aforementioned pathway represents the foundational logical framework of the IRSP, wherein all hospitals are mandated to interface with their internal EHRs strictly according to the prescribed pathway, as illustrated by the solid arrows in Fig 1. Hospitals do not possess the authority to modify this configuration. However, at the interaction interface between the IRSP and EHRs, there exists an "Overlook Access" pathway, indicated by the dashed lines in Fig 1, which hospitals are permitted to configure based on their specific requirements, as the IRSP does not impose explicit guidelines for this component. If an optional 'Overlook Access' pathway unblocked, the physician was permitted to intentionally bypass both the review of reports and the decision-making process, thereby facilitating the immediate execution of local orders. The IRSP forfeits the mutual recognition of decisions regarding the examination/test reports delivered for the physician, with only the delivery record being retained.

In the process of connecting the EHRs of the 8 hospitals with the IRSP, to ensure that the internal structure and clinical specialties of the research subjects were comparable, we randomly selected general hospitals, TCM hospitals, and specialized hospitals respectively to block the "overlook access" pathway. In the end, 2 general hospitals and 1 TCM hospital achieved the pathway blocking, while 1 specialized hospital failed to realize it. The three hospitals were categorized into the intervention group: Hospital A (General), Hospital C (General), and Hospital F (Traditional Chinese Medicine), where the IRSP was implemented, requiring physicians to access and evaluate all delivered reports prior to making decisions. In contrast, three control hospitals—Hospital B (General), Hospital D (General), and Hospital E (Traditional Chinese Medicine)—maintained unrestricted 'Overlook Access' capabilities. The assignment of hospitals was randomized, with two of the four general hospitals and one of the two Traditional Chinese Medicine hospitals allocated to the intervention group, while the remaining hospitals served as controls. Specialty hospitals (G: Cancer, H: Children's) were excluded from the intervention analysis due to incomplete clinical specialty coverage, but they contributed to secondary comparisons. All eight hospitals underwent concurrent monitoring throughout the 12-month study period (from October 1, 2023, to September 30, 2024), ensuring that the disease structures and

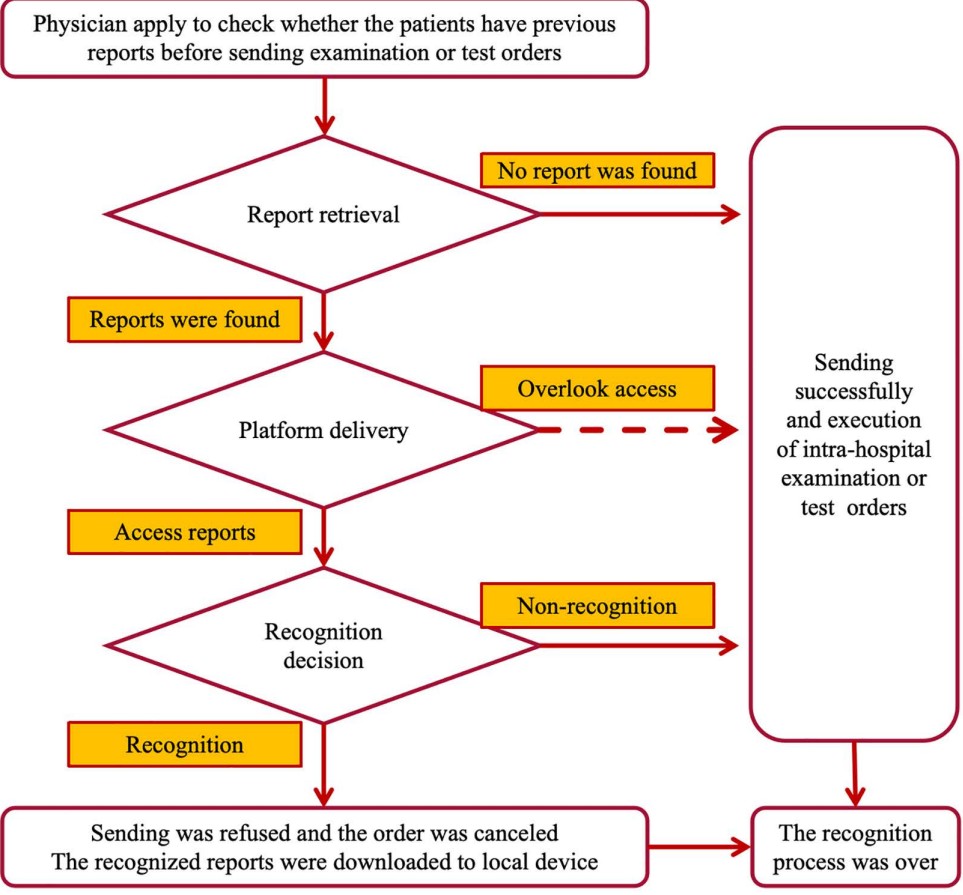

**Fig 1. Workflow of the IRSP with the intervention path highlighted.** This flowchart illustrates the IRSP for physicians ordering examinations/tests, with key steps as follows: Physicians first verify if patients have prior reports during the initial check before issuing new orders; subsequent attempts are made to retrieve existing reports, and if none are found, intra-hospital examination/test orders proceed directly; for found reports, the system delivers them to the platform, where the dashed line ("Overlook access") represents an intervention-blocked path—intervention groups prevent this "overlook access" route; physicians then make a recognition decision, choosing to recognize the retrieved reports (by downloading them locally and canceling new orders) or not recognize them (by proceeding with new orders); the process concludes with the final outcome of either orders being executed (if no reports exist or non-recognition occurs) or canceled (if recognition happens).

patient demographics between the intervention and control groups were comparable, thus mitigating potential confounding effects.

## Recognition metrics

The performance metrics developed for MRP-MER facilitate the monitoring and analysis of mutual recognition behaviors and the subsequent benefits experienced by participating hospitals in the IRSP. These metrics offer a comprehensive reflection of the performance of hospitals, clinical specialties, and physicians within the IRSP framework. They provide a quantitative foundation for in-depth research into the operational effectiveness of the IRSP, thereby assisting hospital management in evaluating the efficiency of IRSP operations, identifying existing issues, and implementing timely, targeted improvement measures. The performance metrics are calculated based on aggregated platform events (accesses, recognitions). The definitions and calculation methods of various metrics are shown in Table 1.

**Table 1. Five of key performance metrics for MRP-MER in IRSP.**

| Performance metrics | Acronym | Definition |
|---|---|---|
| Access Rate | AR | % of delivered reports accessed (reviewed) |
| Cross-Hospital Access Rate | CHAR | % of cross-hospital delivered reports accessed |
| Total Recognition Rate | TRR | % of delivered reports recognized |
| Cross-Hospital Recognition Rate | CHRR | % of cross-hospital delivered reports recognized |
| Overlooked Access Rate | OAR | % of delivered reports overlooked (bypassed) |

Cross-hospital metrics (CHAR, CHRR) specifically measure interactions involving external healthcare institutions; OAR represents the inverse behavior of AR (AR + OAR = 100% for any given report delivery event)

## Statistical analyses

The analyses were organized into three primary dimensions: (1) Intervention Effect: Core recognition metrics (AR, TRR, CHAR, CHRR) were compared between the intervention group (Hospitals A, C, F; n = 3) and the control group (Hospitals B, D, E; n = 3) using magnitude-based inference analysis. Effect sizes (Cohen's d) were calculated as standardized differences between group medians, with 90% confidence intervals derived from 5000 bootstrap resamples to quantify uncertainty. Clinical significance was classified using established thresholds (trivial: $|d| < 0.2$; small: $0.2 \le |d| < 0.5$; moderate: $0.5 \le |d| < 0.8$; large: $|d| \ge 0.8$), with beneficial effects defined as >75% probability of exceeding the minimal clinically important difference threshold (d = 0.2). (2) Clinical Specialty Variation: Associations between medical specialties and recognition metrics (AR, TRR, CHAR, CHRR) were assessed using Spearman's rank correlation coefficients ($\rho$) in the control group (Hospitals B, D, E) to avoid intervention confounding. Specialty-level data aggregated from outpatient departments were ranked ordinally (1–12), with correlation strengths interpreted as: weak ($|\rho| < 0.3$), moderate ($0.3 \le |\rho| < 0.5$), or strong ($|\rho| \ge 0.5$). Significance testing employed permutation-based p-values (10,000 iterations) to account for non-normal distributions and hospital-level clustering. (3) Hospital Type Comparisons: Differences between specialty and general hospitals were assessed for Pediatrics (Specialty Hospital H vs. General Hospitals B, D) and TCM (TCM Hospital E vs. General Hospitals B, D) using descriptive statistics (mean percentages) with 95% confidence intervals presented graphically using error bars.

## Results

### Impact of intervention on recognition metrics

The intervention targeting the 'Overlook access' pathway significantly altered physicians' compliance with the cross-hospital information recognition workflow, as evidenced by changes in core recognition metrics. Specifically, the intervention that blocked the 'Overlook access' pathway reshaped physicians' handling of delivered information, resulting in distinct variations in key metrics between the intervention group (Hospitals A, C, F) and the control group (Hospitals B, D, E) (Fig 2).

In terms of AR, defined as the percentage of delivered reports reviewed, the intervention group exhibited a significantly higher performance (median = 98.9%) compared to the control group (median = 44.6%), demonstrating a large beneficial effect size (Cohen's d = 2.02; 90% CI [1.25, 316.86]). Similarly, CHAR, which represents the access of cross-hospital delivered reports, showed a 41.9% absolute improvement in the intervention group (median = 99.5% vs. 57.6%), yielding the largest effect size (d = 2.85; 90% CI [1.78, 62.37]). Conversely, TRR, which measures the recognition of delivered reports, was paradoxically lower in the intervention group (median = 18.2% vs. 44.3% in controls), exhibiting a large negative effect (d = −2.72; 90% CI [−8.14, −1.25]). Regarding CHRR, which assesses the recognition of cross-hospital reports, minimal between-group differences were observed (intervention median = 49.6% vs. control median = 44.4%; d = 0.40; 90% CI [−2.16, 3.53]), leading to uncertain clinical interpretation.

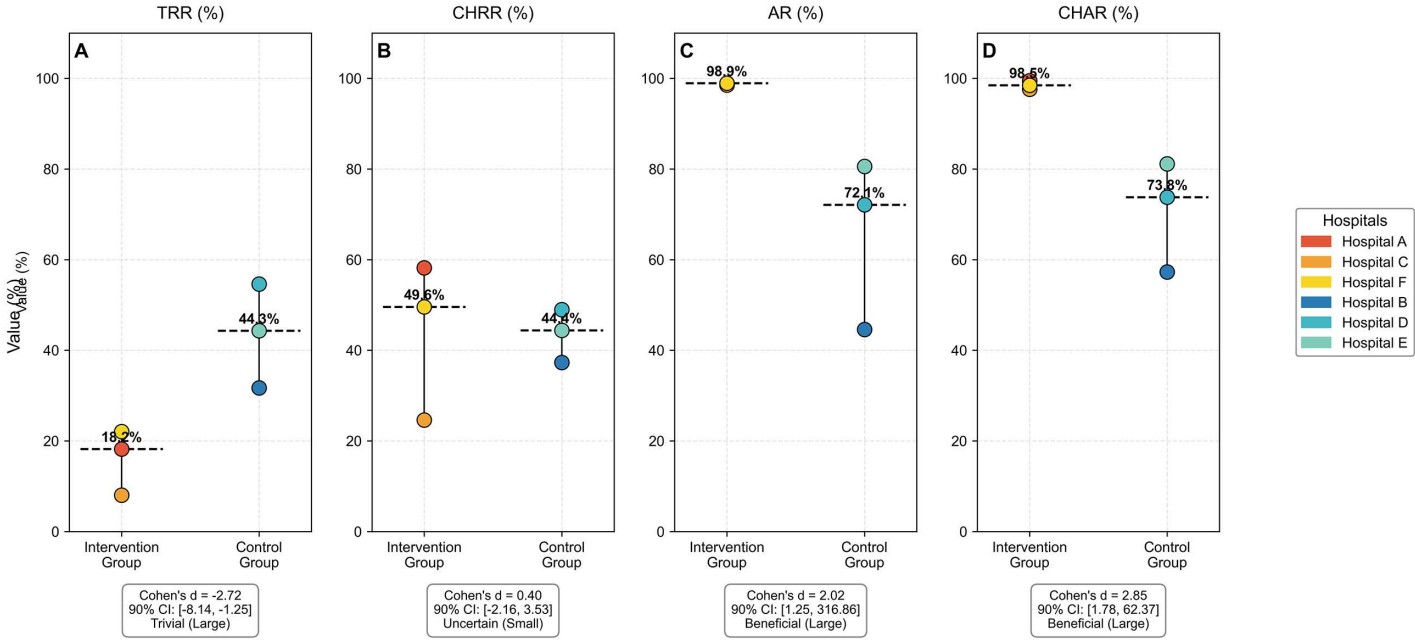

**Fig 2. Comparative analysis of hospital information recognition metrics between intervention and control groups across six hospitals.** Panel A displays Total Recognition Rate (TRR), B shows Cross-Hospital Recognition Rate (CHRR), C presents Access Rate (AR), and D illustrates Cross-Hospital Access Rate (CHAR). Individual data points represent hospital-specific values (warm colors: intervention hospitals A, C, F; cool colors: control hospitals B, D, E), dashed horizontal lines indicate group medians with percentage values labeled, vertical lines show full data ranges, and statistical summaries below each panel report Cohen's d effect sizes with 90% confidence intervals (derived from 5000 bootstrap resamples) alongside clinical significance classifications based on magnitude-based inference thresholds (trivial: $|d| < 0.2$; small: $0.2 \leq |d| < 0.5$; moderate: $0.5 \leq |d| < 0.8$; large: $|d| \geq 0.8$), where beneficial effects indicate >75% probability of exceeding the minimal clinically important difference threshold ($d = 0.2$).

Hospital-level analysis revealed consistent benefits of the intervention in access metrics, with rates exceeding 97.6% compared to 81.1% in the control group. However, recognition patterns varied, indicating that the implementation of the intervention differentially affected information access and documentation workflows. The significant improvements in access metrics suggest a clinically meaningful adoption of the intervention, while the findings related to recognition rates highlight the need for further optimization of processes.

## Clinical specialty variations in mutual recognition

The comprehensive heatmap analysis (Fig 3) reveals substantial variations in healthcare interoperability metrics across 12 clinical specialties and three hospitals, with several key patterns emerging. In terms of AR, Orthopedics maintained the highest overall rate (AR_Total: 83.95%) with strong performance across all hospitals (AR_B: 85.34%, AR_E: 84.71%, AR_D: 78.95%), while Thoracic Surgery (34.42%) and Nephrology (36.83%) showed the lowest overall AR, particularly at Hospital B (30.42% and 21.97%, respectively). For Recognition Rate Disparities, Gastroenterology achieved the highest TRR (TRR_Total: 56.99%), driven largely by Hospital B's exceptional performance (64.59%), whereas Hepatobiliary faced the most significant challenges (TRR_Total: 6.42%), with critically low rates at Hospital B (3.47%); Pediatric care also showed notable variation between hospitals (TRR_B: 35.14% vs TRR_D: 57.51%). In Cross-Hospital Metrics, TCM excelled in both CHAR (CHAR_Total: 83.48%) and CHRR(CHRR_Total: 53.25%), with outstanding results at Hospital D (CHAR_D: 92.99%, CHRR_D: 91.40%), while Endocrinology faced substantial interoperability issues (CHRR_Total: 14.45%) with low inter-hospital recognition rates (CHRR_E: 16.13%, CHRR_D: 20.24%). Hospital-specific patterns included Hospital D's superior performance in Pediatrics (AR_D: 96.35%, CHAR_D: 95.81%) and Traditional Chinese

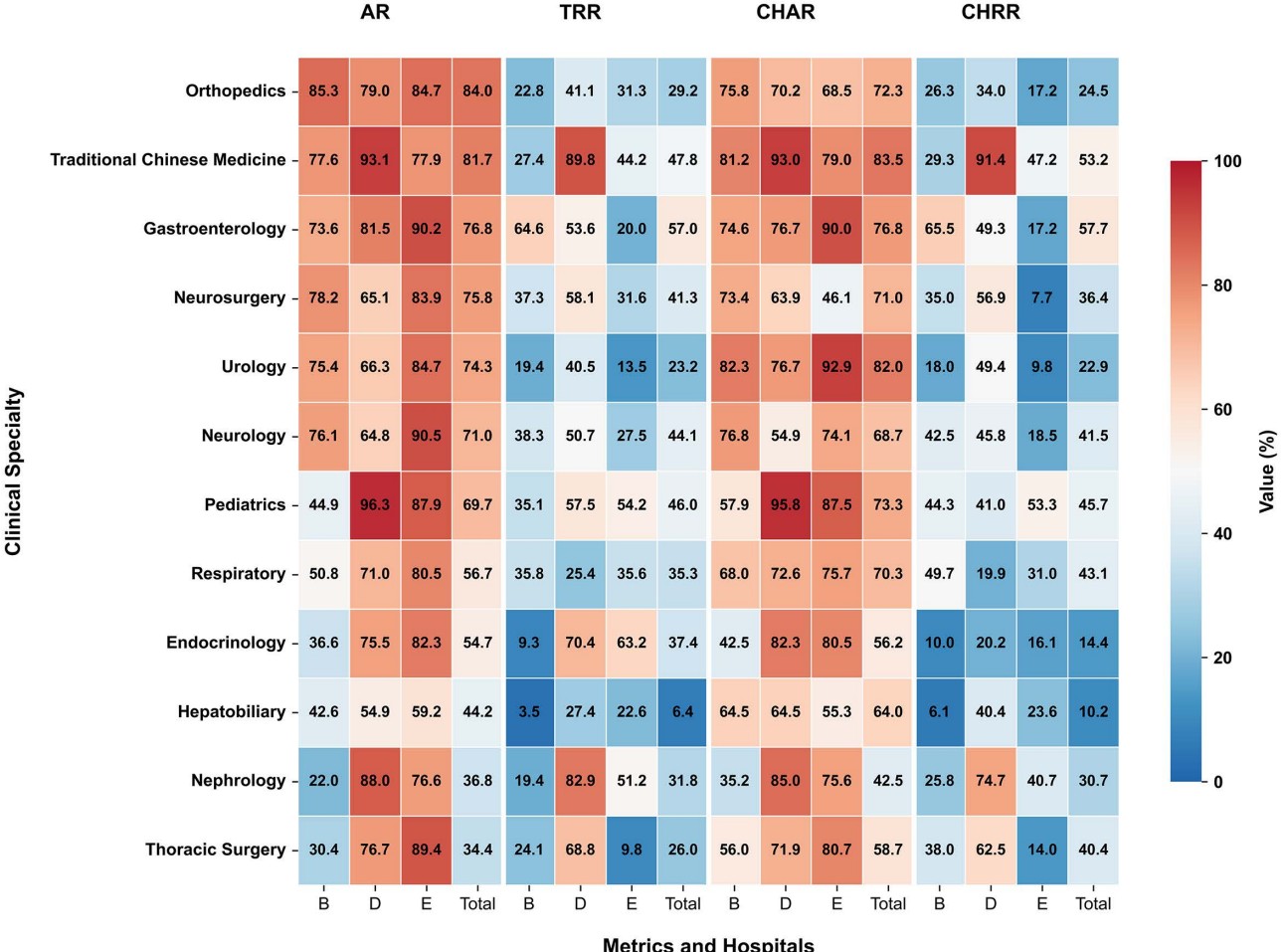

**Fig 3. Performance metrics across clinical specialties and hospitals.** The heatmap illustrates four key metrics: Access Rate (AR), Total Recognition Rate (TRR), Cross-Hospital Access Rate (CHAR), and Cross-Hospital Recognition Rate (CHRR) for 12 clinical specialties across three hospitals (B, D, E) and their combined total. Values represent percentages (0–100%) with a blue-to-red color gradient indicating performance levels.

Medicine (AR_D: 93.07%, CHAR_D: 92.99%); Hospital B's strength in Gastroenterology (TRR_B: 64.59%, CHRR_B: 65.47%) but weakness in Hepatobiliary (AR_B: 42.57%, TRR_B: 3.47%); and Hospital E's excellence in Respiratory care (AR_E: 80.46%, CHAR_E: 75.73%) alongside challenges in Neurosurgery (CHRR_E: 7.69%).

The analysis indicates that while some specialties (e.g., Orthopedics, Traditional Chinese Medicine) maintained consistently high performance across metrics, others (e.g., Hepatobiliary, Endocrinology) faced significant operational challenges.

### Correlations among recognition metrics

The correlation analysis of four mutual recognition metrics across multiple clinical specialties in hospitals is illustrated in Fig 4. There is a strong correlation within access performance metrics (AR and CHAR, ρ = 0.716, p < 0.001) in Panel A and within recognition performance metrics (TRR and CHRR, ρ = 0.733, p < 0.001) in Panel B, In contrast, there is a weak and non-significant relationship between access performance metrics and recognition performance metrics (AR and TRR, ρ = 0.31; CHAR and CHRR, ρ = 0.28) in Panels C and D, which further verifies the aforementioned disconnection.

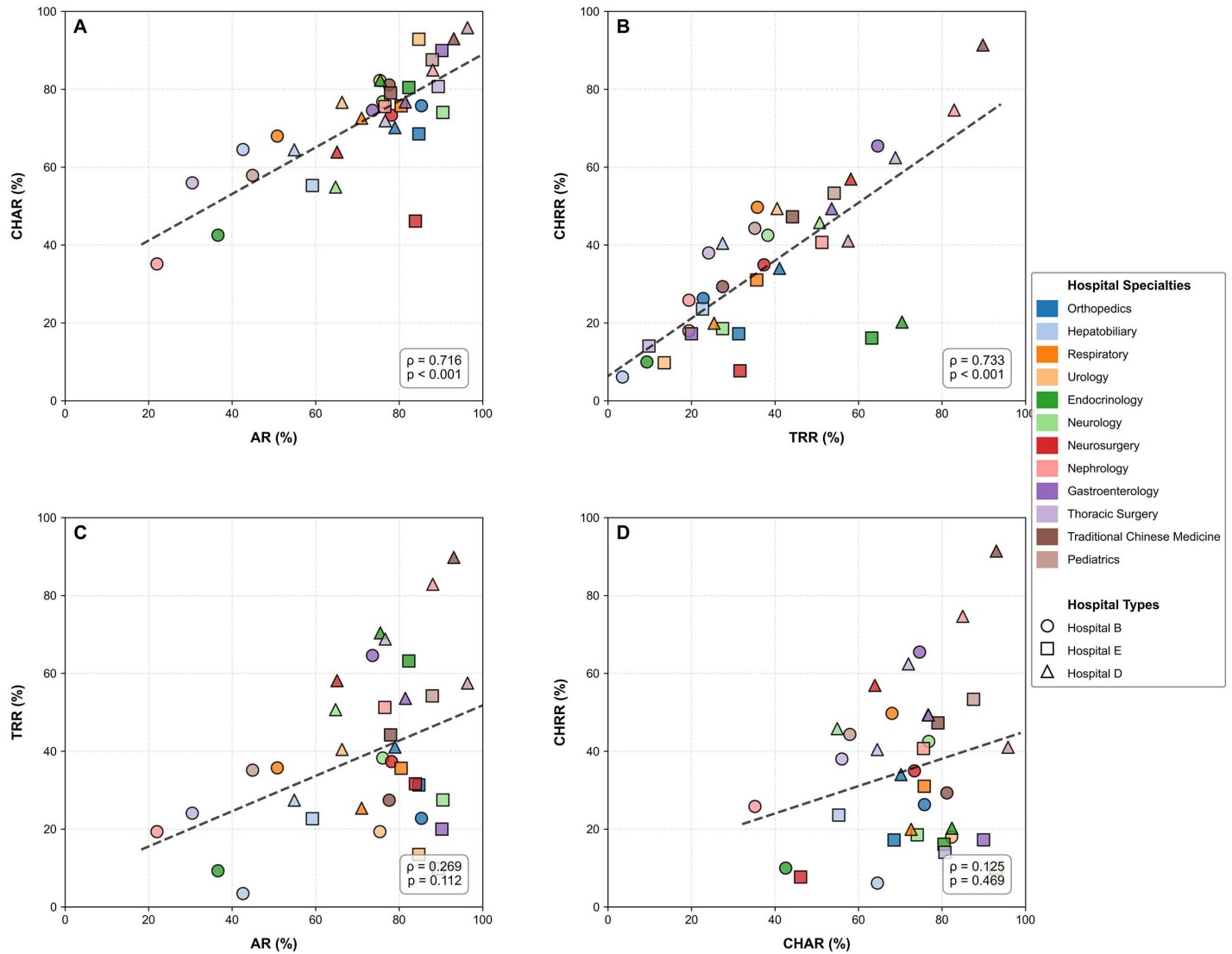

**Fig 4. Metric correlation matrix.** Scatter plots with regression lines depicting correlations between hospital performance metrics across 12 medical specialties. **(A)** Acceptance rate (AR) versus Cross-Hospital Access Rate (CHAR); **(B)** Total Recognition Rate (TRR) versus Cross-Hospital Recognition Rate (CHRR); **(C)** AR versus TRR; **(D)** CHAR versus CHRR. Each point represents one of three hospital types (B, E, D) for a given specialty, with colors distinguishing medical specialties and shapes indicating hospital types (circle: B, square: E, triangle: D**)**. Black dashed lines represent linear regression fits. Spearman's rank correlation coefficients (ρ) and corresponding p-values are reported in the bottom right of each panel, with p < 0.001 indicating statistical significance at α = 0.05. All metrics are expressed as percentages (0-100% scale).

## Access and recognition variations by clinical specialty and hospital management

The results of this study reveal substantial heterogeneity in access rates (AR, CHAR) and recognition rates (TRR, CHRR) across various clinical specialties and hospitals. This finding underscores the stratified influence of clinical domains and institutional management on mutual recognition behaviors, as illustrated in Fig 5.

When stratified by clinical specialty, Pediatrics faced notable challenges in CHRR. Despite achieving high access rates (AR: 96.35%, CHAR: 95.81%), Hospital D exhibited a significantly lower CHRR of 41.03%. Hospital H presented an extreme case, with high access rates (AR: 86.56%, CHAR: 86.17%) but near-negligible recognition rates (TRR: 2.05%, CHRR: 2.82%).

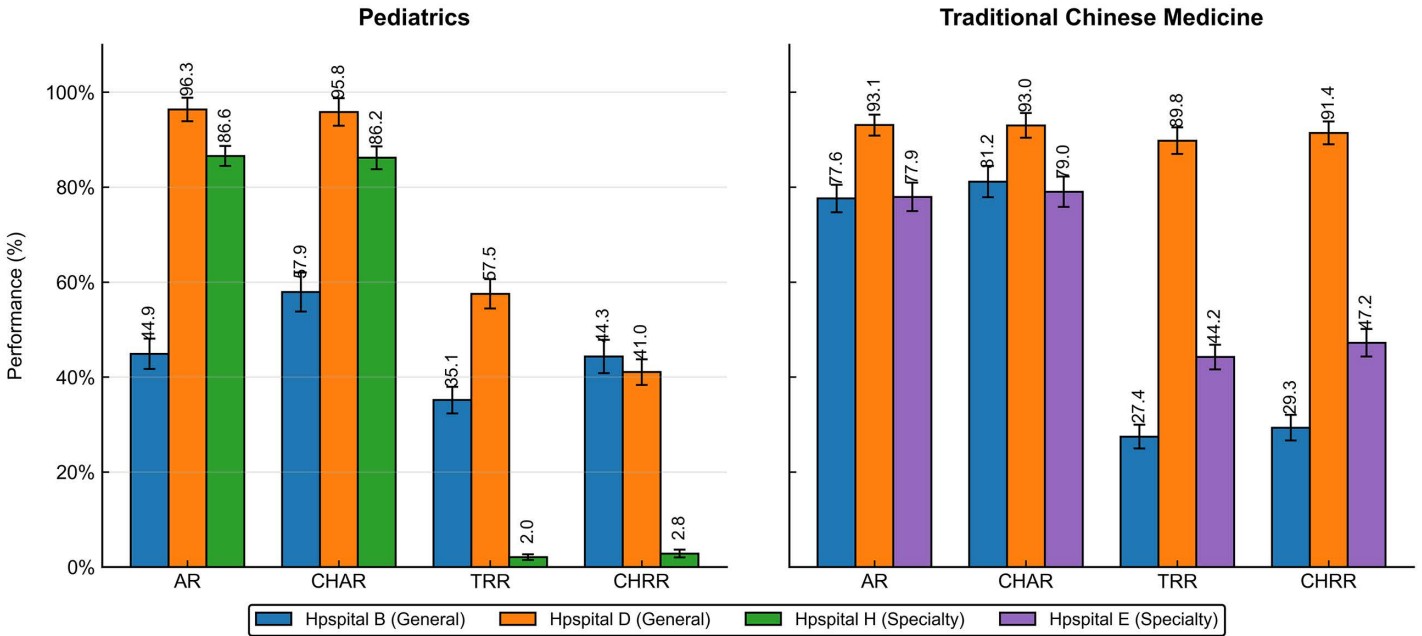

**Fig 5. Comparison of recognition performance metrics on pediatrics and TCM in general and specialty hospitals.** The pediatric hospitals (left panel) include two general hospitals (B, D) and one pediatric specialty-focused hospital (H), while TCM hospitals (right panel) include two general hospitals (B, D) and one TCM specialty-focused hospital (E); metrics evaluated are Access Rate (AR), Cross-Hospital Access Rate (CHAR), Total Recognition Rate (TRR), and Cross-Hospital Recognition Rate (CHRR); data are presented as mean percentages with 95% confidence intervals (error bars).

Hospital B demonstrated moderate access yet low recognition. Insufficient demand-driven factors were identified as barriers to the reuse of external examination reports in pediatrics. Specifically, the physiological status of pediatric patients—particularly young children—undergoes rapid changes, significantly shortening the relevance of historical examination results. This temporal constraint leads clinicians to prefer ordering new tests instead of relying on reports from external institutions. Additionally, certain pediatric conditions, such as acute infections and growth and development assessments, necessitate dynamic and continuous monitoring; thus, isolated instances of external data offer limited clinical reference value, further diminishing the inclination to reuse cross-institutional reports. Challenges related to data applicability also emerged as a key barrier. Pediatric laboratory metrics, including reference ranges for routine blood tests, vary with age, and cross-institutional data may exhibit discrepancies in standardization. These variations increase the cognitive burden on physicians when interpreting external results. Additionally, imaging examinations—such as pediatric bone age assessments—are highly dependent on equipment specifications and operator expertise, which raises concerns about the comparability of external reports and undermines clinician confidence in their reuse. This consistent pattern across hospitals indicates inherent specialty-specific barriers to recognition, possibly arising from the diagnostic complexity, acuity of conditions, or unique therapeutic protocols in pediatrics. Such factors may reduce the reuse of recent examination and test reports generated externally.

In contrast, the TCM generally exhibited a better alignment between access and recognition, particularly in certain hospitals. Hospital D achieved exceptionally high rates across all metrics: AR at 93.07%, CHAR at 92.99%, TRR at 89.78%, and CHRR at 91.40%. Hospital E also demonstrated moderate access with an AR of 77.93% and a CHAR of 79.03%, leading to moderate recognition rates of TRR at 44.20% and CHRR at 47.22%. Only Hospital B displayed a significant access-recognition gap similar to that observed in Pediatrics. This suggests that the standardized principles, diagnostic methods, or treatment modalities within TCM may foster greater trust and acceptance of cross-hospital information when supported by effective hospital systems.

When stratified by hospital management, Hospital D consistently performed well in access metrics (AR/CHAR) across both specialties, achieving over 93% in TCM and over 95% in Pediatrics. This indicates a robust systemic infrastructure, including interoperable IT systems and efficient workflows, that enables clinicians to retrieve external records effectively. However, its recognition rates (TRR/CHRR) varied significantly by specialty, with high rates in TCM (~90%) and moderate to low rates in Pediatrics (~57%/41%). This highlights that while strong hospital management is essential for facilitating access, it alone cannot ensure recognition, which is largely influenced by specialty-specific factors and the perceived reliability and utility of the accessed information within that clinical context. Hospital H (Pediatrics) exemplified a critical situation where efficient access infrastructure (high AR/CHAR:~86%) failed to translate into recognition (very low TRR/CHRR:~2%). This indicates potential systemic issues specific to Hospital H's pediatrics department or its integration with external sources, which severely undermined clinician trust and the usability of the accessed data, outweighing any management efforts to enable access. Hospital B demonstrated moderate access but universally low recognition across both specialties (Pediatrics TRR/CHRR:~35%/44%; TCM TRR/CHRR:~27%/29%). This suggests institutional-level barriers, possibly related to organizational culture, training, incentives, or trust mechanisms, which hinder the utilization of accessed external data regardless of clinical specialty. Hospital E (TCM) achieved moderate access and recognition, indicating reasonably functional management systems that support both access and utilization within the TCM context at this institution, albeit not to the level of Hospital D. These findings collectively emphasize the complex interplay between clinical specialty and hospital management in shaping cross-hospital data recognition, highlighting the need for targeted strategies to address specialty-specific and institutional barriers.The results of performance metricss for each clinical specialty in the hospitals of this study are shown in S1 File.

## Discussion

The implementation of the DRG reform within China's healthcare system has created a conducive environment for the advancement of the MRP-MER [24,25]. This DRG payment reform has catalyzed the development of the MRP-MER, effectively addressing the dual challenges of escalating healthcare costs and the growing demand for high-quality, cost-effective medical services in China. The IRSP, which functions as the data exchange for MRP-MER, facilitates the sharing of medical information among hospitals, enabling physicians to easily access authorized examination and test reports of patients from other healthcare institutions [26,27]. This reform reduces expenses associated with redundant examinations and tests across various levels of hospitals within the hierarchical medical system over a short period. [19,28] In a specific city in China, the DRG reform has notably reduced the standard deviation of hospitalization costs for patients diagnosed with chronic obstructive pulmonary disease, acute myocardial infarction, and cerebral infarction [29]. This reduction can be partly attributed to the effective decrease in unnecessary redundant examinations and tests.Similarly, in a two-month study conducted in an emergency department in western New York State, querying patient history through the HIE was significantly associated with a 52% reduction in the total number of anticipated laboratory tests and a 36% reduction in radiology examinations [30].

We stratified and randomly selected eight large tertiary hospitals from the pilot provinces, which included general hospitals and various specialized facilities, accounting for 70% of all large tertiary hospitals. This selection provides broad coverage and, more importantly, reflects and represents the mutual recognition needs of clinical specialties, thereby highlighting the role of hospital management in this process. Large tertiary hospitals offer a wide range of medical services across various clinical specialties, and their structural similarities enable them to encapsulate the differences among these specialties within a smaller sample size. This approach allows for a detailed examination of the specific mutual recognition needs related to disease diagnosis and treatment across different clinical specialties [31]. Although the mutual recognition rate in large tertiary hospitals is relatively low, several factors contribute to this phenomenon. Firstly, patients admitted to large tertiary hospitals often present with complex medical conditions [32]. Due to fluctuations in patients' health and concerns regarding medical safety, clinicians must meticulously evaluate the usability of examination and test reports from

other institutions. Secondly, the advanced medical technologies utilized in large tertiary hospitals, combined with the intricate nature of patients' conditions and the high demand for diagnostic precision, often render the examination and testing equipment, as well as the quality control standards in primary or lower-level hospitals, inadequate to meet the stringent requirements established by tertiary hospitals [33–35]. Consequently, the examination and test reports issued by these hospitals may not meet the diagnostic and therapeutic needs of large tertiary hospitals. As a result, the mutual recognition rate among large tertiary hospitals may be lower compared to that of primary and secondary hospitals. This discrepancy can serve as a reference baseline for management when evaluating mutual recognition metrics in the regional medical field. In terms of hospital management, large tertiary hospitals are exemplary in their information technology transformation and management practices, often serving as models for lower-level hospitals to emulate. Utilizing large tertiary hospitals as case studies for mutual recognition management research can provide valuable guidance for subsequent mutual recognition efforts in lower-level hospitals.

The findings of this study provide valuable insights into the intricate relationship between access behavior and recognition behavior within the framework of inspection and test reciprocity. Our results clearly indicate that access rates (AR, CHAR) and recognition rates (TRR, CHRR) are governed by fundamentally distinct mechanisms. Technical interventions [36], hospital management practices [37], and clinical specialty characteristics exert unique influences that merit careful consideration. A significant insight from this study is the disconnection between the access process and the recognition process. Interventions aimed at addressing the 'Overlook Access' pathway have significantly improved access metrics, achieving near-perfect levels of 98.80% and 98.53% for AR and CHAR, respectively, in comparison to the control group. This underscores that access is a behavior that can be modified through technological means, suggesting the potential to foster the habit of physicians accessing patients' external medical examination and test reports. Hospitals where physicians actively engage in accessing or recognizing precisely delivered examination or test reports also exhibit high utilization of these reports across hospitals. Therefore, irrespective of the differences among hospitals, measures should be implemented to enhance the AR or TRR for both in-hospital and cross-hospital reports. However, despite the improvement in access, the TRR has subsequently declined, and the improvement in the CHRR is minimal. In the Fig 2, Panels C and D reflect the complexity involved in recognizing previous examination and test reports in clinical diagnosis and treatment concerning access and recognition: a higher AR or CHAR does not directly enhance the TRR or CHRR, as it is further influenced by the specific circumstances and needs of the patient's current episode of diagnosis and treatment. This indicates that recognition is not merely a byproduct of access but a distinct process dependent on specific contexts. There exists a strong correlation within access metrics (AR, CHAR) and within recognition metrics (TRR, CHRR), whereas the relationship between access metrics and recognition metrics is weak and insignificant. This further validates the previously mentioned disconnect between physicians' access and recognition. The capacity to enhance the TRR or CHRR by improving the AR or CHAR through technical information interventions is limited. These findings challenge the notion that merely improving access will naturally lead to better recognition [38,39], emphasizing the necessity to treat recognition (TRR and CHRR) as an independent concept influenced by factors beyond the control of technical workflows, such as alignment with current diagnostic intent. These influences manifest in the dominant role of various clinical specialty characteristics in recognition behaviors, which often outweigh the effects of information interventions. The consistent pattern across clinical specialties, even persisting in control hospitals, highlights one of the primary drivers of specialty-specific workflows, reporting requirements, and diagnostic recognition systems. For instance, orthopedics consistently demonstrates high recognition rates in both TRR and CHRR, likely due to its reliance on objective and enduring data such as imaging, whose clinical utility can be maintained over an extended period [40,41]. For example, X-ray images of fractures remain stable throughout the healing cycle [42,43]. Pediatrics, respiratory medicine, gastroenterology, and other specialties starkly contrast with orthopedics. The test results for disease progression in these fields often change rapidly, making existing results inadequate for reflecting a patient's current disease status [44,45]. These factors diminish the utility of existing test data and complicate correlation assessments. Due to concerns regarding diagnostic and treatment safety,

the TRR or CHRR among these clinical specialties is relatively low. For instance, physiological parameters in pediatrics, such as CRP and bilirubin, have short biological half-lives (typically around 24 hours) [46], necessitating repeated testing by physicians. This situation is not a flaw of the mutual recognition platform but rather a consequence of the data lifecycle dictated by the nature of the diseases involved. Future policies should establish specialty-specific temporal rules (e.g., 'Validity period of pediatric blood routine reports: 3 days') instead of pursuing a uniform recognition rate. Furthermore, we have observed that TCM shows a distinct preference for cross-hospital data over intra-hospital data. This inclination suggests that TCM's diagnostic cognitive system typically emphasizes holistic and cumulative descriptions of patient disease trajectories, potentially benefiting more from comprehensive external data sources than from fragmented intra-hospital reports [47,48]. Such specialty-specific patterns indicate that interventions aimed at improving recognition rates must be tailored to the unique data requirements and clinical reasoning processes of each specialty, rather than relying on generic technological solutions.

The study revealed that, despite the uneven effectiveness of mutual recognition across various specialties within Hospital D, both the access metrics and the recognition metrics surpassed those of other hospitals. Furthermore, the hospital's mutual recognition management strategy significantly influenced the mutual recognition metrics. Through the investigation of Hospital D, it was found that the hospital employed a more targeted approach to mutual recognition management: (1) Training sessions were conducted to familiarize all physicians with the operation of the IRSP system; (2) By enhancing internal performance management of mutual recognition metrics, physicians were encouraged to interpret patients' external examination and test reports without the added pressure of technical information intervention; (3) A campaign was launched within the hospital to promote mutual recognition of medical examinations and tests among both physicians and patients, urging patients to remind physicians of test results obtained from other hospitals. Additionally, physicians can conveniently access patients' reports via IRSP; (4) Resources were allocated to implement specialized technical safeguards, including upgrading the hospital's network bandwidth to 1000 Mbps to ensure rapid access of large imaging files, and maintaining the priority system to reduce the downtime of the interoperability results in IRSP to less than 0.1%. These measures effectively addressed the primary pain points identified in the pre-implementation survey. The management experiences and strategies adopted by Hospital D serve as exemplary cases for other hospitals aiming to enhance the rate of medical mutual recognition in subsequent stages.

Another particularly important aspect is the need to strengthen the management of examination and test quality across different hospitals. Given the perceived quality disparities among hospitals [49–51], the accuracy, consistency, and standardization of examination and test results from medical institutions are crucial considerations for physicians when interpreting reports from other institutions. Ensuring consistent examination and test quality across hospitals is the cornerstone of the MRP-MER. In China, the National Health Commission promotes standardization through several initiatives: Technical Guidelines, which include documents such as the 'Technical Specifications for Medical Imaging Quality Control (2023)' and 'Clinical Laboratory Quality Indicators' aimed at unifying equipment calibration and technical training [52–54]; a hierarchical quality control network, which establishes provincial radiology and laboratory quality control centers (e.g., Zhejiang Center for Clinical Laboratories) to conduct surprise inspections and implement external quality assessment (EQA) [55–58], requiring all hospitals connected to the IRSP to participate in EQA and achieve qualified results; and international practices focusing on certification and interoperability, where the Joint Commission International mandates traceable quality control processes [59], with 98% of accredited hospitals utilizing standardized Laboratory Information Systems and Radiology Information Systems; furthermore, the European Federation of Clinical Chemistry and Laboratory Medicine implements the 'one sample, one result' principle through the EQA Scheme program [60,61], which aims to reduce the inter-laboratory coefficient of variation for key testing items to 15% [58].

## Limitation

This study presents certain limitations, including its exclusive focus on large tertiary hospitals, the effects of variability in hospital management execution, and the omission of other mutual recognition metrics that may influence the five

main metrics. These factors should be considered when interpreting the research findings. To maintain consistency in the clinical specialties offered by the hospitals under investigation and to focus exclusively on large Grade A tertiary hospitals participating in the pilot program, the total number of hospitals included in this research was limited, resulting in a relatively small sample size. This restriction may have constrained the statistical power of some subgroup analyses. Nonetheless, the hospitals examined account for 70% of the large Grade A tertiary hospitals within the pilot program, thus providing a degree of representativeness concerning the mutual recognition situation in the pilot regions. Furthermore, there are notable differences between large Grade A tertiary hospitals and community hospitals regarding doctors' experience levels, the effectiveness of implementing hospital mutual recognition policies, and technical variations in internal EHRs. Our subsequent follow-up study intends to increase the sample size of hospitals and implement information interventions in additional control hospitals to directly compare the impact of these interventions on the mutual recognition rate. Additionally, the disparities among subspecialties within major disciplines (such as pediatric oncology) warrant further investigation. Although baseline comparability between intervention and control group hospitals has been achieved through matching criteria such as hospital size and clinical specialty structure, potential differences beyond these criteria may still exist, potentially affecting the robustness of the comparison results. Despite these limitations, the support from multi-source evidence and rigorous statistical validation provides a solid foundation for the research conclusions. This contributes to addressing the issue of limited sample size and enhances comparability. Furthermore, the mutual recognition performance indicators utilized in mutual recognition management far exceed the five items included in this study. The impact and internal relationships of other mutual recognition performance indicators require further investigation.

## Conclusion

A distinct gap exists between information access and actual recognition of examination and test results; while information technology interventions can substantially improve access to external reports, they do not necessarily lead to higher recognition rates, underscoring the need for tailored strategies targeting recognition itself. Critically, recognition behavior is highly dependent on clinical specialty-specific factors, as the clinical utility of external data varies markedly among fields such as orthopedics, pediatrics, and TCM. These variations reflect differences in data stability, temporal validity, and fundamental diagnostic approaches. Furthermore, institutional success hinges upon the adoption of effective, multifaceted policies designed to improve recognition rates. The IRSP serves as an exemplar of China's progress in pragmatic interoperability, substantially reducing redundant testing. Nevertheless, these findings collectively highlight that sustainable improvement in recognition relies on aligning external data with clinical relevance and decision-making processes, rather than simply increasing the accessibility of examination results.

## Supporting information

**S1 File. Showing performance metrics outcomes for all clinical departments in the study hospitals.**
(XLSX)

## Acknowledgments

The authors would like to express their sincere gratitude to all the individuals and organizations that contributed to the successful completion of this study.Special thanks go to the participating hospitals for their cooperation and for the valuable insights they provided into the practical aspects of mutual recognition practices. The authors extend their sincere gratitude to the leadership, information technology departments, and clinical staff of all participating hospitals in Zhejiang Province for their invaluable cooperation, data sharing, and insights into the practical implementation of the IRSP and mutual recognition workflows. Their engagement was essential for conducting this real-world evaluation. We also

acknowledge the Zhejiang Provincial Health Commission and the administrators of the IRSP their role in advancing the MRP-MER policy.

## Author contributions

**Conceptualization:** Chao Song, Shasha Qian, Shuning Liu, Jun Zhou.

**Data curation:** Chao Song.

**Formal analysis:** Chao Song, Xinmian Huang, Jun Zhou.

**Funding acquisition:** Chao Song, Shuning Liu.

**Investigation:** Chaoyun Yuan, Shuning Liu.

**Methodology:** Chao Song, Xinmian Huang.

**Project administration:** Chao Song, Chaoyun Yuan.

**Resources:** Chao Song, Shasha Qian, Chaoyun Yuan.

**Software:** Xinmian Huang, Chaoyun Yuan.

**Supervision:** Chao Song, Shasha Qian, Chaoyun Yuan, Jun Zhou.

**Validation:** Chao Song.

**Writing – original draft:** Chao Song, Jun Zhou.

**Writing – review & editing:** Chao Song, Xinmian Huang, Shuning Liu, Jun Zhou.

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
