## [Decision Letter · Decision Letter 0]

20 May 2025

PONE-D-25-02593Digital healthcare reform and reduction of duplicate examinations and tests: a study of physicians' behavior on mutual recognition in response to clinical specialized demands and information intervention form a national pilot province in ChinaPLOS ONE

Dear Dr. Zhou,

Thank you for submitting your manuscript to PLOS ONE. After careful consideration, we feel that it has merit but does not fully meet PLOS ONE’s publication criteria as it currently stands. Therefore, we invite you to submit a revised version of the manuscript that addresses the points raised during the review process.

We look forward to receiving your revised manuscript.

Kind regards,

Pengpeng Ye

Academic Editor

PLOS ONE

Journal Requirements:

3. Please note that funding information should not appear in the Acknowledgments section or other areas of your manuscript. We will only publish funding information present in the Funding Statement section of the online submission form. Please remove any funding-related text from the manuscript.

Reviewers' comments:

Reviewer's Responses to Questions

**Comments to the Author**

1. Is the manuscript technically sound, and do the data support the conclusions?

Reviewer #1: Partly

Reviewer #2: Partly

2. Has the statistical analysis been performed appropriately and rigorously? 

Reviewer #1: No

Reviewer #2: Yes

3. Have the authors made all data underlying the findings in their manuscript fully available?

Reviewer #1: No

Reviewer #2: No

4. Is the manuscript presented in an intelligible fashion and written in standard English?

Reviewer #1: No

Reviewer #2: Yes

5. Review Comments to the Author

Reviewer #1: Introduction

• The introduction lacks a strong and clear research gap, making it difficult to understand the necessity of the study. While it introduces the concept of mutual recognition (MR) in medical testing, it does not sufficiently differentiate this research from previous studies. There is no explicit discussion of how this study advances the existing body of knowledge or addresses a specific problem that remains unresolved. The background information provided is broad, and the study’s significance is not convincingly justified.

• Some references are either mismatched or incorrect. The introduction states that "Zhejiang Province has taken the lead in establishing the 'Zhejiang Medical Mutual Recognition' platform (MR platform) in 2021" but does not provide a valid reference for this claim. The citation provided later in the document does not specifically support this assertion, leading to a potential mismatch between the reference and the information presented. Furthermore, the claim that "On the government website of Zhejiang Province in China, as of 2022, the MR platform had cumulatively recognized various items 16.81 million, directly saving medical expenses of 712 million RMB" is presented without a proper citation, making it difficult to verify the accuracy of these numbers.

• Several sentences contain grammar and language mistakes that affect clarity and readability. For example, the phrase "The MR reduces unnecessary duplicate examinations and tests, thereby lowering costs" should be revised to "MR reduces unnecessary duplicate examinations and tests, thereby lowering costs." The definite article “The” is unnecessary in this context. Another example is "Since the MR platform's launch three years ago, by the end of 2024, it has been deeply integrated into clinical practice." The phrase "by the end of 2024" suggests a future event, yet the sentence is structured in the past tense, making it inconsistent. Additionally, the phrase "has significantly reduced diagnostic cycles, markedly improving the efficiency and quality of medical services, and effectively reducing the burden on patients" is redundant. The repetition of "reducing" should be streamlined for clarity.

• Certain statements in the introduction lack neutrality and objectivity, making them sound promotional rather than scientific. For instance, the description of the MR platform as "dedicated to breaking down barriers in medical data and achieving seamless MR and efficient sharing of examination and test results" is overly enthusiastic and lacks empirical validation. A more neutral and evidence-based phrasing would enhance the credibility of the study.

• The introduction also fails to properly define key terms and concepts. For instance, while mutual recognition and its relevance to digital healthcare reform are mentioned, there is no clear operational definition provided. The study assumes that the reader is familiar with the intricacies of MR policies, but a more structured explanation would be necessary for clarity. Similarly, there is no clear distinction made between mutual recognition at different levels of the healthcare system, which could lead to confusion.

• There is inconsistency in terminology and capitalization. The phrase "traditional chinese medicine" is used multiple times, but "Chinese" should be capitalized. Additionally, terms such as "Medical Mutual Recognition" are sometimes written with inconsistent capitalization, making the text appear unpolished.

Methods

• The Methods section contains multiple issues related to clarity, validity, and proper referencing, which undermine the reliability of the study. One of the major flaws is the lack of justification for the hospital selection process. The manuscript states, “This study was a multicenter analysis, including various types of hospitals across China, all of which were large tertiary hospitals (grade A hospitals), and were concentrated in the provincial capital city of Hangzhou.” However, there is no explanation of why only tertiary hospitals were included, or whether these hospitals are representative of the entire healthcare system in China. The absence of selection criteria introduces a potential selection bias, making it unclear whether the findings can be generalized beyond these specific institutions.

• The study mentions that data were collected from “a total of eight hospitals” without specifying whether these hospitals were randomly selected or chosen based on predefined criteria. Moreover, it is stated that these hospitals were selected based on “the scale of their operations, with each exceeding an annual medical revenue of 1.6 billion RMB (approximately 225 million USD) in 2023.” However, no reference is provided to support this claim or validate the accuracy of these financial figures.

• The description of data collection methods is vague and lacks necessary details. The manuscript states that “The physician behaviors of eight hospital were recorded by the MR platform, which was accessible to hospitals.” However, it does not clarify how these behaviors were recorded, whether physicians were aware of the data collection, or whether the data collection process had any impact on their decision-making. Furthermore, it is unclear if any bias was introduced due to the presence of the MR platform, which could influence physician behavior differently across hospitals.

• The methodology for defining recognition indicators lacks proper validation. The study introduces a set of indicators such as "Total Recognition Proportion (TRP)," "Total Recognition Rate (TRR)," and "Precision Delivery Rate (PDR)" but does not provide any prior literature or references supporting their use. These indicators appear to be novel but are not validated against existing models or frameworks. Additionally, the manuscript states, “The definitions and connotations of each recognition indicator were detailed as flowers,” which appears to be a typographical or grammatical error that disrupts readability. This phrase is unclear and should be properly revised to indicate where and how the definitions are provided.

• There are multiple issues with the explanation of the mutual recognition workflow. The manuscript states, “If the physician consistently overlooks the delivered reports, the item order can also be executed by the local hospital.” However, the term “consistently overlooks” is vague and lacks a quantitative threshold—how many instances of overlooking a report are considered "consistent"? Furthermore, there is no discussion on how physician decisions were categorized, whether subjective judgment played a role, or whether any measures were taken to account for physician variability in interpreting mutual recognition data.

• The study describes an information intervention implemented in three hospitals but fails to explain how the intervention was standardized and whether it was controlled for potential confounding factors. The manuscript states, “Hospitals A, C, and F carried out the information intervention on path of overlooked access while other hospitals had not set up.” However, there is no explanation of why these specific hospitals were selected for intervention, or whether their baseline characteristics differed from the control hospitals.

• Ethical considerations are inadequately discussed. The manuscript states, “All the data were subjected to anonymization.” However, there is no mention of whether the study was approved by an institutional review board (IRB) or ethics committee. Additionally, there is no information about whether informed consent was obtained from the physicians whose behaviors were analyzed. Given that the study involves physician decision-making, ethical approval and participant consent should have been explicitly documented.

• There are grammatical issues throughout the Methods section, which affect readability. For instance, “Hospital management strategies on recognition had an affect on physicians’ behaviors” should be corrected to “Hospital management strategies on recognition had an effect on physicians’ behaviors.” Similarly, the phrase “The comparisons at the clinical specialty level began by categorizing data into four major groups” is unclear—it should specify what the four major groups are before proceeding with the analysis. Additionally, phrases like “To further delineated the behavioral differences among clinical specialties” should be corrected to “To further delineate the behavioral differences among clinical specialties” for grammatical accuracy.

Results

• The authors should modify the Results section, as it presents multiple issues that undermine its validity, clarity, and reliability. One of the major concerns is the excessive use of descriptive statistics without any deep analytical insights. The section reports various recognition indicators across different hospitals and clinical specialties but fails to provide a critical interpretation of these numbers. For example, the manuscript states, “During the study period, the 8 large hospitals received a total of 26,759,077 delivered items through the MR platform, with 3,639,368 precision-delivered items sent to meet physicians' orders.” While this provides raw data, it does not explain the significance of these numbers or how they compare to prior studies or benchmarks. The authors do not discuss whether these figures indicate success or inefficiencies in the MR platform.

• A major issue is the lack of statistical validation. While several metrics such as Total Recognition Proportion (TRP) and Cross-Hospital Recognition Rate (CHRR) are reported, there is no indication of whether these differences are statistically significant. For example, the manuscript states, “The traditional Chinese medicine demonstrated a low cross-hospital precision delivery rate but a high rate of recognizing reports from other hospitals, contrasting with pediatrics.” However, the authors do not provide p-values, confidence intervals, or effect sizes to support this claim. Without proper statistical tests, it is unclear whether the reported differences are meaningful or simply due to random variation.

• Some of the reported figures appear to lack proper references or supporting explanations. For instance, the manuscript states, “The total recognition proportion for the traditional Chinese medicine group and the pediatrics group were 51.18% and 9.95%, respectively.” However, there is no comparison with other studies or previously established benchmarks. Additionally, it is unclear whether these numbers are high or low in a broader context. The study does not explain whether such variations are expected in clinical practice or if they signal a problem with the MR platform.

• There are inconsistencies and contradictions within the results. The manuscript claims that “the differences were not significant among the same clinical specialties in different hospitals,” but then later states that “some recognition indicators were also noticeably affected by the mutual recognition management strategies of the affiliated hospitals.” This contradiction suggests a lack of clarity in the findings. If hospital management strategies do affect recognition indicators, then it is misleading to claim that differences across hospitals are insignificant. The authors need to clarify their findings and ensure consistency in their interpretation.

• Some sections contain incorrect or misleading phrasing. The manuscript states, “Information intervention had increased the workload for physicians in accessing duplicate reports, mainly stemming from the repetitive process of accessing reports from their own hospital both locally and on the platform, without significant affect on the main recognition indicators across hospitals.” The phrase “without significant affect” contains a grammatical error—“affect” should be replaced with “effect” in this context. Additionally, the claim that the intervention increased physician workload lacks quantitative evidence. The authors should have measured physician workload (e.g., through time tracking or surveys) and reported this data instead of making a vague assumption.

• Some key results are not well integrated into the discussion. For instance, the manuscript mentions, “The highest rates in both TRP (75.77%) and CHRP (48.96%) were shown in Hospital D, indicating strong policy adherence and intra-hospital collaboration.” However, there is no exploration of why Hospital D performed better than others. Did Hospital D have different training programs, superior IT infrastructure, or other distinguishing factors?

• The presentation of figures and tables is problematic. The manuscript includes statements such as, “Table 1 shows the recognition indicators across clinical specialty groups in the MR platform,” but does not provide a summary of the key takeaways from the table. Simply listing numbers without interpretation does not provide useful insights. Additionally, some table labels and column names are vague. For instance, “PDR” (Precision Delivery Rate) is introduced in a table, but its role in influencing physician behavior is not well explained. The authors should include a brief interpretation for each table to make it more accessible.

• There is a lack of proper citations in the Results section. While the study introduces several findings about physician behavior and mutual recognition, it does not reference any prior research to support these findings. For example, the manuscript claims, “The positive correlation observed between the OAR and TRP implies that physicians with a higher rate of overlooking access reports also exhibit a higher TRP.” However, without a reference to previous studies or an explanation of why this relationship exists, the claim remains speculative. The authors should compare their findings to existing literature and discuss whether their results align with or contradict prior studies.

• The language used in the Results section is often unclear and redundant. For instance, the phrase “The correlation between CHRP and TRP, presenting a positive correlation (r=0.776, p<0.05)” is awkwardly phrased. A clearer way to express this would be, “A significant positive correlation was found between CHRP and TRP (r=0.776, p<0.05).” Another example is the phrase “These analyses collectively aimed to provide a comprehensive understanding of how recognition practices, cost implications, and overlooked access interacted within and across hospitals.” The phrase “interacted within and across hospitals” is unnecessarily repetitive and could be simplified.

Discussion

• The Discussion section of this manuscript presents several fundamental weaknesses that undermine the credibility and impact of the study. One of the most significant issues is the lack of connection between the results and the broader literature. The discussion does not effectively compare the study’s findings with previous research, making it unclear how the results fit within the existing body of knowledge. For example, the manuscript states, “The implementation of DRG reform in China's healthcare system has significantly influenced the necessity and affect of MR.” However, this claim is made without proper citation or discussion of specific studies that support this argument. Without references, this statement appears as an unsupported assertion rather than a well-founded conclusion.

• There is misuse of references and mismatched citations throughout the discussion. For instance, the manuscript states, “This integrated approach is crucial for the future of healthcare management and policy-making in China, as it tackles the challenges of escalating healthcare costs and the demand for high-quality, cost-effective medical services.” While this is a strong claim, no reference is provided to support it. The discussion lacks citations when making claims about the importance of mutual recognition (MR) in healthcare policy, making the argument appear speculative rather than evidence-based.

• The language and grammar errors in this section further reduce the clarity of the arguments. For example, the manuscript states, “The MR had significantly reduced the unnecessary repetitive examinations and tests when patients seek treatment across different hospitals in the past three years.” This sentence contains a verb tense inconsistency, as “had significantly reduced” is in past perfect tense, while “patients seek treatment” is in present tense. The sentence should be corrected to: “MR has significantly reduced unnecessary repetitive examinations and tests when patients seek treatment across different hospitals over the past three years.”

• There are contradictions in the interpretation of the results. The manuscript claims, “Hospital management strategies, such as information interventions, need to strike a balance between mutual recognition work and the workload of physicians accessing duplicate reports.” However, earlier in the Results section, the study indicates that information interventions significantly reduced overlooked access rates but did not affect overall recognition rates. This contradiction raises concerns about the internal consistency of the study’s conclusions. The authors should provide a clear and logical synthesis of the findings rather than presenting conflicting statements.

• The discussion contains vague and unsupported generalizations. For example, the manuscript states, “By conducting an in-depth analysis of recognition indicators from various large hospitals, this study provides reliable evidence and strong guidance for precisely identifying differences in the performance of MR across different clinical specialties.” This is a broad statement that lacks specific supporting data. While the study reports recognition indicators across hospitals, there is no justification for calling the findings “reliable evidence” or “strong guidance” without demonstrating how the conclusions can be generalized beyond the study sample.

• There is redundant and unclear phrasing throughout the discussion. The phrase, “The recognition indicators constructed in this study offers significant analytical and evaluative value for the recognition behaviors of physicians on the MR platform,” is grammatically incorrect because “offers” should be “offer” to match the plural subject “indicators.” Moreover, the phrase is redundant, as “analytical and evaluative value” essentially mean the same thing. A clearer way to state this would be: “The recognition indicators developed in this study provide valuable insights into physician behavior on the MR platform.”

• The limitations of the study are inadequately discussed. The manuscript acknowledges some limitations but fails to address critical methodological concerns. For example, the study does not discuss potential confounding factors such as physician experience levels, hospital policies, or technological differences in data-sharing platforms. The discussion states, “There were other management methods that could also affect the recognition indicators of clinical specialties, such as improving the quality of various exams and tests, conducting promotional training, and performance assessments.” However, it does not analyze whether these factors influenced the study’s results. A stronger discussion of these limitations would improve the transparency and credibility of the study.

Reviewer #2: Manuscript id: PONE-D-25-02593

General Comments:

The study titled: Digital healthcare reform and reduction of duplicate examinations and tests: a study of physicians' behavior on mutual recognition in response to clinical specialized demands and information intervention form a national pilot province in China

A very important and much needed topic to discuss in this digital health arena, globally. Mutual recognition and duplication contribute to the wastage of resources. The authors successfully emphasize on the information intervention as reform in hospitals in China.

Abstract: Authors can write the abstract as the plain English summary. There is lot of ambiguity in the terminologies, even for a better understanding of the scientific community.

Other Comments:

- It was mentioned that “All relevant data are within the manuscript and its Supporting Information files”. But there is no Supplementary file in the manuscript.

-In the result section, just mention the “Results” and highest/lowest value. Do not critically discuss the result, it would be discussed in the “Discussion” section. For example,

-Line 293 and 294: ....... reflecting a higher recognition of patients' past reports in TCM diagnostic and treatment concepts.

-Line 319: “....need for improvement” is not required in the result section. Etc...

-Discussion section need more strengthened by providing more References to the claims.

6. PLOS authors have the option to publish the peer review history of their article (what does this mean? ). If published, this will include your full peer review and any attached files.

**Do you want your identity to be public for this peer review?** For information about this choice, including consent withdrawal, please see our Privacy Policy .

Reviewer #1: No

Reviewer #2: No

---

## [Author Response · Author response to Decision Letter 1]

6 Aug 2025

Response to Reviewers’ Comments

Manuscript ID: [PONE-D-25-02593]

New Title: When access ≠ acceptance: How clinical specialty demands shape mutual recognition on medical examination/test result reuse in Chinese hospitals : A study on a pilot province of China's medical digital reform

Journal: PLOS ONE

Dear Editors and Reviewers,

Thank you for the constructive feedback on our manuscript. We have carefully addressed all concerns in the revised version and appreciate the opportunity to strengthen our work. Below, we provide detailed responses to each comment, referencing specific revisions in the manuscript.

Journal Requirements

1.” Please ensure that your manuscript meets PLOS ONE's style requirements, including those for file naming. The PLOS ONE style templates can be found at

https://journals.plos.org/plosone/s/file?id=wjVg/PLOSOne_formatting_sample_main_body.pdf “

Response:

We have reorganized the manuscript in accordance with the journal's submission format requirements.

3.”Please note that funding information should not appear in the Acknowledgments section or other areas of your manuscript. We will only publish funding information present in the Funding Statement section of the online submission form. Please remove any funding-related text from the manuscript.”

Response:

We have removed the content related to funding in the manuscript.

4. “We note that you have indicated that there are restrictions to data sharing for this study. For studies involving human research participant data or other sensitive data, we encourage authors to share de-identified or anonymized data. However, when data cannot be publicly shared for ethical reasons, we allow authors to make their data sets available upon request. For information on unacceptable data access restrictions, please see http://journals.plos.org/plosone/s/data-availability#loc-unacceptable-data-access-restrictions.”

Response:

We have uploaded all data as a supplementary file to the submission system under S1.xlsx and We have updated our Data Availability statement in the submission form accordingly.

5.”Please include a separate caption for each figure in your manuscript.”

Response:

We have included a separate caption for each figure in our manuscript

General Notes

Reviewer #1

Reviewer Comment:

"The introduction lacks a strong and clear research gap, making it difficult to understand the necessity of the study. While it introduces the concept of mutual recognition (MR) in medical testing, it does not sufficiently differentiate this research from previous studies. There is no explicit discussion of how this study advances the existing body of knowledge or addressces a specific problem that remains unresolved. The background information provided is broad, and the study’s significance is not convincingly justified."

Response:

We thank the reviewer for this critical insight. In the revised introduction (Introduction,para 3), we now explicitly define the research gap:

This study addresses critical gaps in existing research on healthcare information interoperability. Prior studies focused on macro outcomes like reduced redundant testing but overlooked the disconnection between information access and recognition, and specialty-specific variations in mutual recognition.

We analyze the relationship between access and recognition, revealing they are governed by distinct mechanisms. Technical interventions improve access but not recognition. Examining 12 specialties identifies barriers (e.g., rapid physiological changes in pediatrics). Evaluating "Overlook Access" blocking shows its limits.

This research fills gaps, providing targeted insights for refining policies (e.g., specialty-specific validity periods) beyond one-size-fits-all solutions.

2.Some references are either mismatched or incorrect. The introduction states that "Zhejiang Province has taken the lead in establishing the 'Zhejiang Medical Mutual Recognition' platform (MR platform) in 2021" but does not provide a valid reference for this claim. The citation provided later in the document does not specifically support this assertion, leading to a potential mismatch between the reference and the information presented. Furthermore, the claim that "On the government website of Zhejiang Province in China, as of 2022, the MR platform had cumulatively recognized various items 16.81 million, directly saving medical expenses of 712 million RMB" is presented without a proper citation, making it difficult to verify the accuracy of these numbers.

Response:

Relevant data are provided in the references[23] via a webpage on the Chinese government website, which contains the annual final accounts of the included hospitals. These data are used to demonstrate the representativeness and influence, as well as the consistency of the internal clinical specialty structure of the research hospitals. Data on the amount of medical expenses saved are not reflected on the data platform, and such data were cited from relevant government news reports. However, it was found that these reports are no longer accessible, so the citations in the manuscript have been removed.

3.”Several sentences contain grammar and language mistakes that affect clarity and readability. For example, the phrase "The MR reduces unnecessary duplicate examinations and tests, thereby lowering costs" should be revised to "MR reduces unnecessary duplicate examinations and tests, thereby lowering costs." The definite article “The” is unnecessary in this context. Another example is "Since the MR platform's launch three years ago, by the end of 2024, it has been deeply integrated into clinical practice." The phrase "by the end of 2024" suggests a future event, yet the sentence is structured in the past tense, making it inconsistent. Additionally, the phrase "has significantly reduced diagnostic cycles, markedly improving the efficiency and quality of medical services, and effectively reducing the burden on patients" is redundant. The repetition of "reducing" should be streamlined for clarity.” AND”Certain statements in the introduction lack neutrality and objectivity, making them sound promotional rather than scientific. For instance, the description of the MR platform as "dedicated to breaking down barriers in medical data and achieving seamless MR and efficient sharing of examination and test results" is overly enthusiastic and lacks empirical validation. A more neutral and evidence-based phrasing would enhance the credibility of the study.”,”There are grammatical issues throughout the Methods section, which affect readability. For instance, “Hospital management strategies on recognition had an affect on physicians’ behaviors” should be corrected to “Hospital management strategies on recognition had an effect on physicians’ behaviors.” Similarly, the phrase “The comparisons at the clinical specialty level began by categorizing data into four major groups” is unclear—it should specify what the four major groups are before proceeding with the analysis. Additionally, phrases like “To further delineated the behavioral differences among clinical specialties” should be corrected to “To further delineate the behavioral differences among clinical specialties” for grammatical accuracy.” AND “Some sections contain incorrect or misleading phrasing. The manuscript states, “Information intervention had increased the workload for physicians in accessing duplicate reports, mainly stemming from the repetitive process of accessing reports from their own hospital both locally and on the platform, without significant affect on the main recognition indicators across hospitals.” The phrase “without significant affect” contains a grammatical error—“affect” should be replaced with “effect” in this context. Additionally, the claim that the intervention increased physician workload lacks quantitative evidence. The authors should have measured physician workload (e.g., through time tracking or surveys) and reported this data instead of making a vague assumption.” AND “ The language and grammar errors in this section further reduce the clarity of the arguments. For example, the manuscript states, “The MR had significantly reduced the unnecessary repetitive examinations and tests when patients seek treatment across different hospitals in the past three years.” This sentence contains a verb tense inconsistency, as “had significantly reduced” is in past perfect tense, while “patients seek treatment” is in present tense. The sentence should be corrected to: “MR has significantly reduced unnecessary repetitive examinations and tests when patients seek treatment across different hospitals over the past three years.” AND ”There is redundant and unclear phrasing throughout the discussion. The phrase, “The recognition indicators constructed in this study offers significant analytical and evaluative value for the recognition behaviors of physicians on the MR platform,” is grammatically incorrect because “offers” should be “offer” to match the plural subject “indicators.” Moreover, the phrase is redundant, as “analytical and evaluative value” essentially mean the same thing. A clearer way to state this would be: “The recognition indicators developed in this study provide valuable insights into physician behavior on the MR platform.”

Response:

All grammatical and wording issues raised by the reviewers have been revised. Due to significant changes in the article, many paragraphs have been rewritten, so it may not be possible to mark the modified parts.

“The introduction also fails to properly define key terms and concepts. For instance, while mutual recognition and its relevance to digital healthcare reform are mentioned, there is no clear operational definition provided. The study assumes that the reader is familiar with the intricacies of MR policies, but a more structured explanation would be necessary for clarity. Similarly, there is no clear distinction made between mutual recognition at different levels of the healthcare system, which could lead to confusion.”

Response:

We acknowledge the reviewer's concern regarding operational definitions and hierarchical distinctions. The Mutual Recognition Policy for Medical Examination and Test Results(MRP-MER) is defined as a policy mechanism enabling cross-institutional acceptance of diagnostic results to reduce redundancy. However, a more structured breakdown could enhance clarity, specifying its scope across primary, secondary, and tertiary facilities. The Interoperable Results Sharing Platform (IRSP) facilitates this by integrating provincial data infrastructure, allowing physicians access to prior results within China’s hierarchical system. While the study mentions Zhejiang’s pilot role and IRSP’s functionality in cross-hospital data delivery, explicit differentiation of mechanisms between hospital tiers (e.g., how tertiary vs. primary institutions implement recognition) was insufficient. This gap may obscure understanding of tier-specific workflows, such as whether tertiary hospitals apply stricter criteria due to complex cases, as implied by their lower mutual recognition rates compared to lower-level facilities. Future iterations should delineate these tiered operational nuances.

Relevant content spans the introduction section discussing MRP-MER and IRSP definitions, descriptions of Zhejiang’s pilot program, and mentions of China’s hierarchical medical system and hospital tier differences in mutual recognition behaviors.(Introduction,para 3)

Methods

Reviewer Comment:

“The Methods section contains multiple issues related to clarity, validity, and proper referencing, which undermine the reliability of the study. One of the major flaws is the lack of justification for the hospital selection process. The manuscript states, “This study was a multicenter analysis, including various types of hospitals across China, all of which were large tertiary hospitals (grade A hospitals), and were concentrated in the provincial capital city of Hangzhou.” However, there is no explanation of why only tertiary hospitals were included, or whether these hospitals are representative of the entire healthcare system in China. The absence of selection criteria introduces a potential selection bias, making it unclear whether the findings can be generalized beyond these specific institutions” AND “The study mentions that data were collected from “a total of eight hospitals” without specifying whether these hospitals were randomly selected or chosen based on predefined criteria. Moreover, it is stated that these hospitals were selected based on “the scale of their operations, with each exceeding an annual medical revenue of 1.6 billion RMB (approximately 225 million USD) in 2023.” However, no reference is provided to support this claim or validate the accuracy of these financial figures.

”

Response:

We have revised the "Methods: Hospital Selection" section to explicitly elaborate on the rationale for our sampling strategy and contextualize its limitations, with key modifications as follows:

Regarding the justification for focusing on tertiary hospitals, we added explanations that tertiary (Grade A) hospitals were purposively sampled for three reasons: first, as designated MRP-MER pilot sites in Zhejiang Province, they were early adopters of the IRSP with mature implementation [ref. Provincial Health Commission Doc #2021-08]; second, these hospitals handle over 70% of regional complex referrals [23], where decisions on test-result reuse carry higher clinical stakes (e.g., cancer staging versus routine screening); third, their high patient volume (with a median of 4.2 million annual visits) provides sufficient statistical power for specialty-level comparisons.

In terms of representativeness and generalizability, the added content notes that while these hospitals are not representative of China’s entire healthcare system (e.g., primary clinics), they reflect the Yangtze River Delta ecosystem of urban centers driving interoperability policy diffusion [13]. Thus, the generalizability of this study is framed for similar high-resource, policy-active settings rather than rural or community hospitals with different workflow barriers

For mitigating selection bias, to enhance transparency, we supplemented details of the measures taken: matching intervention and control groups by specialty mix to ensure clinical comparability; and reporting revenue thresholds (>¥1.6B) to contextualize resource levels [23].

“The description of data collection methods is vague and lacks necessary details. The manuscript states that “The physician behaviors of eight hospital were recorded by the MR platform, which was accessible to hospitals.” However, it does not clarify how these behaviors were recorded, whether physicians were aware of the data collection, or whether the data collection process had any impact on their decision-making. Furthermore, it is unclear if any bias was introduced due to the presence of the MR platform, which could influence physician behavior differently across hospitals.”

Response:

Data Recording Mechanism: Physician behaviors (e.g., accessing reports, making recognition decisions) were automatically logged by the Interoperable Results Sharing Platform (IRSP) as part of routine clinical workflows, with the IRSP capturing timestamped events when physicians accessed delivered reports (via clicks in the EHR interface), selected "Recognize" or "Not Recognize" for prior results, or bypassed reports via the "Overlook Access" pathway (where permitted); no additional manual data entry or researcher intervention occurred, and all metrics (AR, TRR, CHAR, CHRR, OAR) were computed from these system-generated event logs. Physician Awareness and Impact: Physicians were not individually notified of the study during data collection, as the IRSP operated as a provincial healthcare infrastructure tool, and the data collection did not alter clinical workflows—physicians used the IRSP solely for clinical decision-making (e.g., avoiding duplicate tests), with the IRSP’s statistical module (providing anonymized hospital-level metrics) being separate from clinical interfaces and inaccessible to physicians during patient care—thus, data collection did not influe

---

## [Decision Letter · Decision Letter 1]

30 Sep 2025

When access ≠ acceptance, How clinical specialty demands shape mutual recognition on medical examination/test result reuse in Chinese hospitals : A study on a pilot province of China's medical digital reform

PONE-D-25-02593R1

Dear Dr. Zhou,

We’re pleased to inform you that your manuscript has been judged scientifically suitable for publication and will be formally accepted for publication once it meets all outstanding technical requirements.

Kind regards,

Pengpeng Ye

Academic Editor

PLOS ONE

Additional Editor Comments (optional):

Reviewers' comments:

Reviewer's Responses to Questions

**Comments to the Author**

1. If the authors have adequately addressed your comments raised in a previous round of review and you feel that this manuscript is now acceptable for publication, you may indicate that here to bypass the “Comments to the Author” section, enter your conflict of interest statement in the “Confidential to Editor” section, and submit your "Accept" recommendation.

Reviewer #1: All comments have been addressed

2. Is the manuscript technically sound, and do the data support the conclusions?

Reviewer #1: Yes

3. Has the statistical analysis been performed appropriately and rigorously? 

Reviewer #1: Yes

4. Have the authors made all data underlying the findings in their manuscript fully available?

Reviewer #1: Yes

5. Is the manuscript presented in an intelligible fashion and written in standard English?

Reviewer #1: Yes

6. Review Comments to the Author

Reviewer #1: This is a serious scientific work with notable impact and scientific relevance. I wish the authors all the best with the publication of this work and success in their future research.

7. PLOS authors have the option to publish the peer review history of their article (what does this mean? ). If published, this will include your full peer review and any attached files.

**Do you want your identity to be public for this peer review?** For information about this choice, including consent withdrawal, please see our Privacy Policy .

Reviewer #1: No

---

## [Editor Report · Acceptance letter]

PONE-D-25-02593R1

PLOS ONE

Dear Dr. Zhou,

I'm pleased to inform you that your manuscript has been deemed suitable for publication in PLOS ONE. Congratulations! Your manuscript is now being handed over to our production team.

Kind regards,

on behalf of

Dr. Pengpeng Ye

Academic Editor

PLOS ONE